# Factors associated with a history of treatment interruption among pregnant women living with HIV in Malawi: A cross-sectional study

**Simone A. Sasse**[1,2]*, **Bryna J. Harrington**[2,3], **Bethany L. DiPrete**[2,4], **Maganizo B. Chagomerana**[2], **Laura Limarzi Klyn**[2], **Shaphil D. Wallie**[2], **Madalitso Maliwichi**[2], **Allan N. Jumbe**[2], **Irving F. Hoffman**[2,5], **Nora E. Rosenberg**[2,6], **Jennifer H. Tang**[2,7], **Mina C. Hosseinipour**[2,5], **on behalf of the S4 Study**[¶]

1 Department of Obstetrics and Gynecology, New York University, New York, New York, United States of America, 2 University of North Carolina Project-Malawi, Kamuzu Central Hospital, Lilongwe, Malawi, 3 Department of Gynecology and Obstetrics, Johns Hopkins University, Baltimore, Maryland, United States of America, 4 Department of Epidemiology, The University of North Carolina at Chapel Hill, Chapel Hill, United States of America, 5 Department of Medicine, The University of North Carolina at Chapel Hill, Chapel Hill, United States of America, 6 Department of Health Behavior, The University of North Carolina at Chapel Hill, Chapel Hill, United States of America, 7 Department of Obstetrics and Gynecology, The University of North Carolina at Chapel Hill, Chapel Hill, United States of America

¶ Membership of the S4 Study Group is listed in the Acknowledgments.
* simone.sasse@nyulangone.org

## Abstract

### Introduction

Long-term care engagement of women on antiretroviral therapy (ART) is essential to effective HIV public health measures. We sought to explore factors associated with a history of HIV treatment interruption among pregnant women living with HIV presenting to an antenatal clinic in Lilongwe, Malawi.

### Methods

We performed a cross-sectional study of pregnant women living with HIV who had a history of ART interruption presenting for antenatal care. Women were categorized as either retained in HIV treatment or reinitiating care after loss-to-follow up (LTFU). To understand factors associated with treatment interruption, we surveyed socio-demographic and partner relationship characteristics. Crude and adjusted prevalence ratios (aPR) for factors associated with ART interruption were estimated using modified Poisson regression with robust variance. We additionally present patients' reasons for ART interruption.

### Results

We enrolled 541 pregnant women living with HIV (391 retained and 150 reinitiating). The median age was 30 years (interquartile range (IQR): 25–34). Factors associated with a history of LTFU were age <30 years (aPR 1.46; 95% CI: 1.33–1.63), less than a primary school education (aPR 1.25; CI: 1.08–1.46), initiation of ART during pregnancy or breastfeeding (aPR 1.49, CI: 1.37–1.65), nondisclosure of HIV serostatus to their partner (aPR 1.39, CI:

**Data Availability Statement:** The data relevant to this study are uploaded to the figshare repository at

DOI: https://doi.org/10.6084/m9.figshare.19425734.v1.

**Funding:** This research was made possible through funding by the National Institutes of Health (R01HD080485, D43TW010060, MH, URL: https://www.nih.gov), the Doris Duke Charitable Foundation International Clinical Research Fellows Program (SAS, URL: https://www.ddcf.org), the UNC Medical Scientist Training Program (T32GM008719, BJH), the NIMH individual fellowship (F30MH111370, BJH, URL: https://www.nimh.nih.gov), the Fulbright-Fogarty U.S. Student fellowship (BJH, URL: https://us.fulbrightonline.org), and the NIH Fogarty International Center (R25TW009340, BJH, URL: https://www.fic.nih.gov). Regulatory support was provided through the UNC Center for AIDS Research (P30AI50410).

**Competing interests:** The authors have declared that no competing interests exist.

1.24–1.58), lack of awareness of partner's HIV status (aPR 1.41, CI: 1.27–1.60), and no contraception use at conception (aPR 1.60, CI 1.40–1.98). Access to care challenges were the most common reasons reported by women for treatment interruption (e.g., relocation, transport costs, or misplacing health documentation).

## Conclusions

Interventions that simplify the ART clinic transfer process, facilitate partner disclosure, and provide counseling about the importance of lifelong ART beyond pregnancy and breastfeeding should be further evaluated for improving retention in ART treatment of women living with HIV in Malawi.

## Introduction

Malawi, a country with an HIV prevalence of 9%, has achieved significant gains in access to HIV testing and antiretroviral therapy (ART) [1]. In 2011, Malawi pioneered Option B+, a prevention of mother to child transmission (MTCT) program whereby pregnant and breastfeeding women living with HIV receive lifelong ART regardless of CD4 count or HIV clinical stage [2]. The national estimate of mother-to-child transmission in Malawi five years after implementation of Option B+ was 3.7% (95% CI 2.3–6.0) among HIV-exposed infants aged 4–12 weeks [3], compared to 8.5% (95% CI 6.6–10.7) in infants under three months early after adoption of Option B+ [4].

Despite these gains (which still fall short of WHO elimination of MTCT criteria of less than 50 infant infections per 100,000 live births [5]), sustained engagement in HIV care remains a challenge, particularly among women enrolled in Option B+, who have increased odds of loss to follow-up (LTFU) from HIV care as compared to women who initiate ART for their own health based on CD4 count or WHO HIV clinical stage. Malawi Ministry of Health (MOH) data reveal that as many as 23.2% of Option B+ women were LTFU one year after ART initiation [6]. Additional factors associated with women's LTFU in Malawi during the early Option B+ era include lower knowledge about the benefits of ART [7], younger age [6, 7], lower level of education [7], and early challenges with adherence [6]. Partner dynamics, such as the fear of HIV disclosure to their partners, are modifiable factors to improve women's engagement in care, but are less studied. Gendered barriers are likely contributors to care engagement challenges in women, who may fear that disclosure could lead to the loss of economic support or intimate partner violence [8–10].

Women living with HIV must consistently take ART for maximal individual [11] and public health [3, 12] benefits. Nearly all women (95%) in Malawi receive antenatal care (ANC) [1]. To determine factors associated with interruption of ART in a more mature Option B+ program, we performed a cross-sectional comparison of characteristics between pregnant women retained in HIV care versus those with a history of LTFU who were reinitiating care. Determining factors associated with ART interruption can help identify potential targets to improve care engagement initiatives for women living with HIV in Malawi.

## Methods

### Study setting and population

Recruitment of participants occurred at two government antenatal clinics (Bwaila Hospital and Area 18 Health Center) in Lilongwe, Malawi from 2015–2019 as a part of an observational

study to evaluate the safety and efficacy of Option B+ (ClinicalTrials.gov identifier NCT02249962). For this study, women who presented for an antenatal clinic visit and tested positive for HIV during any trimester of pregnancy were approached for recruitment into either: 1) the retained cohort, or 2) the reinitiating cohort. Participants were recruited into the retained cohort from June 2015 to November 2016 at Bwaila Hospital if they were on first-line ART, had presented to all clinic visits for at least six months, and were virally suppressed at enrollment per Malawi MOH guidelines (<1000 copies/mL) [13]. In the reinitiating cohort, recruited from June 2015 to March 2019 at Bwaila Hospital and the Area 18 Health Center, eligible participants had previously initiated first-line ART, but were not on ART at enrollment and had not taken ART for at least three weeks at the time of study enrollment, confirmed through either 1) verbal report of discontinuing first-line treatment, 2) documentation of frequently missed appointments at the health facility, or 3) documentation of HIV-positive status while pregnant or breastfeeding after July 2011 (when the implementation of Option B + began).

In addition to testing HIV positive, eligibility criteria included age 16 years or older, intent to give birth in Lilongwe, and capacity and willingness to provide informed consent. Study nurses conducted the consent process and enrollment interviews with participants in Chichewa, the predominant local language. All participants were continued or reinitiated on first-line ART treatment (tenofovir/lamivudine/efavirenz, TDF/3TC/EFV) at enrollment in line with MOH guidelines [13], and provided with intensive ART adherence counseling.

## Measures

Data collected at enrollment included patient demographics, commute time to the clinic, pregnancy history, partner information, WHO HIV clinical status, and HIV testing and treatment history. Baseline blood samples were collected for viral load (VL) and CD4 count. An undetectable VL was defined as <40 copies/mL per the lower limit of detection of the Abbott m200rt PCR system used in the study. In the reinitiating cohort, study nurses asked participants to provide an explanation for discontinuation in free form.

Our primary outcome was a history of ART interruption versus a history of HIV treatment retention. Possible predictor variables were selected based on literature review and indicators of partner dynamics. Continuous and categorical variables were dichotomized as follows: age (<30 vs ≥30 years), gestational age at enrollment ($1^{st}$ or $2^{nd}$ trimester vs $3^{rd}$ trimester), marital status (unmarried vs married), education (none or some primary vs completed primary or above), employment status (unemployed vs employed), commute time to clinic from home residence (<1 hour vs ≥1 hour), intended current pregnancy (yes vs no), any contraceptive method use at conception of current pregnancy (yes vs no), HIV serostatus disclosure to her partner (yes vs no), and awareness of her partner's HIV status (yes vs no). In addition, the type of ART regimen and reason for prior ART initiation was defined for the most recent regimen (the current regimen in the retained group or the regimen prior to stopping care in the reinitiating group). The type of regimen was defined as first-line treatment (TDF/3TC/EFV) versus any other regimen, while the reason for prior ART initiation was classified as either: 1) prevention of MTCT (PMTCT), meaning initiation during pregnancy or breastfeeding as part of Option B+, or 2) own health, meaning initiation outside of pregnancy or breastfeeding. Regimen dates for HIV treatment were ascertained based on health documentation or self report if required. Prior to June 2016, HIV patients were initiated on ART for their own health based on HIV clinical stage or CD4 count per Malawi MOH national guidelines [13], after which Malawi transitioned to a 'test and treat' strategy for all HIV patients.

**Table 1. Characteristics of study participants at enrollment (N = 541).**

| Characteristic | n (%) |
|---|---|
| Age (years) | |
| 16–29 | 264 (49) |
| $\geq$ 30 | 277 (51) |
| Gestational Age | |
| Trimester 1 or 2 | 372 (69) |
| Trimester 3 | 169 (31) |
| Marital Status | |
| Unmarried | 49 (9) |
| Married | 492 (91) |
| Education | |
| None or some primary | 271 (50) |
| Completed primary or higher | 270 (50) |
| Employment | |
| Unemployed | 345 (64) |
| Employed | 196 (34) |
| Commute to Clinic | |
| <1 hour | 181 (33) |
| $\geq$ 1 hour | 360 (67) |
| Intended Current Pregnancy | |
| No | 405 (75) |
| Yes | 136 (25) |
| Use of Contraception[a] | |
| No | 476 (88) |
| Yes | 65 (12) |
| Disclosure of HIV Status to Partner | |
| No | 76 (14) |
| Yes | 465 (86) |
| Aware of Partner's HIV Status | |
| No | 143 (26) |
| Yes | 398 (74) |
| Intimate Partner Violence[b] | |
| No | 449 (83) |
| Yes | 92 (17) |
| ART Regimen Type[c] | |
| TDF/3TC/EFV | 520 (96) |
| Other[d] | 19 (4) |
| Reason for ART Initiation[c] | |
| PMTCT | 250 (46) |
| Own Health | 290 (54) |
| WHO HIV Clinical Stage | |
| Stage 1 | 450 (83) |
| Stage 2–4 | 91 (17) |

[a]Use of any method of contraception at the conception of the current pregnancy.

[b]Verbal or physical intimate partner violence within the last three months, by the sex partner of the current pregnancy.

[c]Missing in 1 retained cohort participant.

[d]Other regimens included stavudine/lamivudine/nevirapine (d4T/3TC/NVP), zidovudine/lamivudine/nevirapine (AZT/3TC/NVP), tenofovir/lamivudine/nevirapine (TDF/3TC/NVP), tenofovir/lamivudine/lopinavir/ritonavir (TDF/3TC/LPV/r), and stavudine/lamivudine/lopinavir/ritonavir (d4T/3TC/LPV/r).

### Analytic methods

**Quantitative.** To explore potential factors associated with ART interruption, we used modified Poisson regression models with robust variance to calculate unadjusted prevalence ratios (PR). Modified Poisson regression with robust variance estimates were utilized due to issues with convergence of log-binomial regression models [14, 15]. Potential predictors of ART interruption with a predetermined p-value of ≤0.20 were selected for an initial multivariable model. Variance inflation factor analyses were performed to ensure lack of collinearity. Next, the model was simplified through an iterative process of manual backward elimination of the variable with the highest p-value until each remaining variable had a p-value of ≤0.05. We present adjusted PRs (aPR) and 95% confidence intervals (CI) of the variables included in the final model.

We describe the duration of treatment in the retained and reinitiating cohorts, using a Mann-Whitney test to compare durations of treatment. We also describe the length of time off treatment in the reinitiating cohort.

**Qualitative.** Qualitative free responses for ART discontinuation in the reinitiating cohort were organized using thematic content analysis [16]. One researcher reviewed all participant responses to the question and developed eight categories for coding. Two independent researchers applied the codebook to the responses. The initial inter-rater reliability Cohen's kappa was 0.86. Discrepancies were reviewed and resolved by consensus. The frequency of reason for ART discontinuation category was tabulated and reported.

All analyses were conducted in STATA version 15.1 [17].

### Ethics approval and consent to participate

The study protocol and consent documents were approved by the institutional review boards of the University of North Carolina at Chapel Hill and the National Health Science Research Committee of the Malawian Ministry of Health. All women signed or fingerprinted a consent form prior to study participation.

## Results

At total of 541 pregnant women living with HIV were recruited. For the retained cohort, 1194 women living with HIV and on treatment were identified at the Bwaila antenatal clinic from March 2015-November 2016. Study nurses approached 79% (n = 945) of these women and enrolled 41% of those approached (n = 391). For the reinitiating cohort, study staff approached 168 women found to have stopped HIV treatment at antenatal care, and 77% (n = 130) of these women enrolled. Twenty additional women in the reinitiating cohort were enrolled from the Area 18 district health center. Due to a lack of data, we were unable to determine the percentage of women approached and enrolled at the Area 18 site.

In the study sample of 541 women across both cohorts (391 retained, 150 reinitiating), the median age was 30 years (interquartile range (IQR): 25–34). Ninety-one percent (n = 492/545) were married, 83% (n = 450/541) presented at WHO Clinical Stage 1, 86% (n = 465/541) had disclosed their HIV status to the sex partner of their current pregnancy, 74% (n = 398/541) were aware of their partner's HIV status, and 46% (n = 250/541) had initiated ART for PMTCT (additional characteristics in Table 1).

A majority (99%) of participants' regimen dates were ascertained through health documentation (self report of regimen dates was utilized in 7/541 participants). Women in the retained sample had been on treatment for a median of 29 months (IQR 22–39, n = 3 missing) prior to enrollment, while the reinitiating cohort had been on treatment for median of 18 months (IQR 5–36, n = 1 missing) prior to LTFU and off treatment for a median of 7 months (IQR 4–16)

prior to enrollment. Within the reinitiating cohort, women starting treatment for PMTCT took ART for a median of 24 months (IQR 7–38) before LTFU, while women starting for their own health took ART for a median of 7 months (IQR 3–34) before LTFU; the distributions of the two groups differed significantly (Mann-Whitney U = 2001, $n_1$ = 93, $n_2$ = 56, p<0.018).

Median CD4 count at enrollment was 539 cells/mm$^3$ (IQR 410–690; 2 missing) in the retained cohort and 394 cells/mm$^3$ (IQR 244–544; 9 missing) in the reinitiating cohort. Amongst the retained women, 96% (n = 376/391) had an undetectable VL, and the median detectable VL was 240 copies/mL (IQR 91–692). In the reinitiating cohort, 8% (n = 12/150) had undetectable VL at enrollment, and the median detectable viral load was 21,850 copies/ mL (IQR 5,102–80,300).

In unadjusted analyses, we found that factors associated with ART interruption included younger age (uPR = 1.51; 95% CI = 1.39–1.68), being unmarried (uPR = 1.35; 95% CI = 1.15– 1.64), intention of the current pregnancy (uPR = 1.53; 95% CI = 1.16–2.02), no use of contraception at the time of conception (uPR = 1.64; 95% CI = 1.46–2.01), lack of serostatus disclosure to their partner (uPR = 1.50; 95% CI = 1.38–1.66), lack of knowledge of their partner's status (uPR = 1.49; 95% CI = 1.37–1.64), and previously beginning ART for PMTCT as opposed to for their own health (uPR = 1.48; 95% CI = 1.35–1.65, Table 2). We found no difference between retained and reinitiating women with respect to gestational age at presentation, education, employment, commute time to clinic, experience of intimate partner violence in the last three months, and type of most recent ART regimen.

Our final multivariable model included age, education, contraception use, HIV disclosure to partner, awareness of partner's HIV status, and reason for ART initiation (Table 2). We found that factors associated with ART interruption were younger age (aPR 1.46; 95% CI: 1.33–1.63), lower education level (aPR 1.25; CI: 1.08–1.46), no contraception use at the time of conception (aPR 1.60, CI: 1.40–1.98), lack of HIV status disclosure to the sex partner of the pregnancy (aPR 1.39, CI: 1.24–1.58), lack of awareness of the sex partner's HIV status (aPR 1.41, CI: 1.27–1.60), and beginning ART during pregnancy or breastfeeding as opposed to her own health (aPR 1.49, CI: 1.37–1.65).

In the reinitiating cohort, eight categories of reasons for ART interruption were identified (Fig 1). The most commonly reported reason for LTFU was a health facility access issue (48%, n = 69/143, Fig 1), mainly due to relocation, transport difficulties to the clinic, or loss of health documentation. The other seven reported reasons for LTFU were ART side effects (13%, n = 19/143), partner challenges (13%, n = 18/143), feeling healthy enough not to require drugs (10%, n = 15/143), fear of stigma from health care workers or family (8%, n = 12/143), mental health difficulties (3%, n = 5/143), religious reasons (3%, n = 4/143), and not having the time to return to clinic (1%, n = 1/143).

## Discussion

We recruited a cohort of pregnant women living with HIV seeking antenatal care in Lilongwe, Malawi in order to explore reasons for interruption of ART. We found that factors associated with ART interruption in women reinitiating ART as compared to women retained in HIV care included: prior initiation of ART for PMTCT rather than for their own health, younger age, lower education status, non-disclosure of their HIV status to their male partner, lack of awareness of their partner's HIV status, and lower rates of contraception use at the time of conception. The most common reasons cited for ART discontinuation included access to health clinics (either due to relocation, transportation costs, or losing health documentation) and treatment side effects. Our findings highlight barriers to engagement and can inform strategies for improving engagement in HIV care in women.

**Table 2. Factors associated with ART interruption among Malawian women presenting for antenatal care.**

| Characteristic | Retained (N = 391) | Reinitiating (N = 150) | Prevalence Ratio | |
|---|---|---|---|---|
| | n (%) | n (%) | Unadjusted (95% Confidence Interval) | Adjusted (95% Confidence Interval) |
| Age | | | | |
| 16–29 | 165 (42) | 99 (66) | 1.51 (1.39–1.68) | 1.46 (1.33–1.63) |
| ≥ 30 | 226 (58) | 51 (34) | 1 | 1 |
| Gestational Age | | | | |
| Trimester 1 or 2 | 274 (70) | 98 (65) | 1 | |
| Trimester 3 | 117 (30) | 52 (35) | 1.17 (0.88–1.55) | |
| Marital Status | | | | |
| Unmarried | 29 (7) | 20 (13) | 1.35 (1.15–1.64) | |
| Married | 362 (93) | 130 (87) | 1 | |
| Education | | | | |
| None or some primary | 186 (48) | 85 (57) | 1.23 (1.04–1.48) | 1.25 (1.08–1.46) |
| Primary or higher | 205 (52) | 65 (43) | 1 | 1 |
| Employment | | | | |
| Unemployed | 242 (62) | 103 (69) | 1.20 (1.00–1.48) | |
| Employed | 149 (38) | 47 (31) | 1 | |
| Commute to Clinic | | | | |
| <1 hour | 136 (35) | 45 (30) | 1 | |
| ≥ 1 hour | 255 (65) | 105 (70) | 1.17 (0.87–1.58) | |
| Intended Pregnancy | | | | |
| No | 306 (78) | 99 (66) | 1 | |
| Yes | 85(22) | 51 (34) | 1.53 (1.16–2.02) | |
| Use of Contraception[a] | | | | |
| No | 333 (85) | 143 (95) | 1.64 (1.46–2.01) | 1.60 (1.40–1.98) |
| Yes | 58 (15) | 7 (5) | 1 | 1 |
| Disclosed HIV status to Partner | | | | |
| No | 39 (10) | 37 (25) | 1.50 (1.38–1.66) | 1.39 (1.24–1.58) |
| Yes | 352 (90) | 113 (75) | 1 | 1 |
| Aware of Partner's HIV Status | | | | |
| No | 81 (21) | 62 (41) | 1.49 (1.37–1.64) | 1.41 (1.27–1.60) |
| Yes | 310 (79) | 88 (58) | 1 | 1 |
| Intimate Partner Violence[b] | | | | |
| No | 328 (84) | 121 (80) | 1 | |
| Yes | 63 (16) | 29 (19) | 1.17 (0.83–1.64) | |
| Reason for ART Initiation[c] | | | | |
| PMTCT | 156 (40) | 94 (63) | 1.48 (1.34–1.66) | 1.49 (1.37–1.65) |
| Own Health | 234 (60) | 56 (37) | 1 | 1 |

[a]Use of any method of contraception at the conception of current pregnancy.

[b]Experience of verbal or physical intimate partner violence within the last three months, by the sex partner of the current pregnancy.

[c]Missing in 1 retained cohort participant.

Programs such as Option B+ have been successful in recruiting women into treatment at the time of pregnancy or breastfeeding; yet there remain a significant number of women who are lost from care following ART initiation for PMTCT as compared to for their own health. In our sample, 47% (n = 250/541) had initiated ART for PMTCT, and starting ART for PMTCT as opposed to for their own health was associated with ART interruption. Our

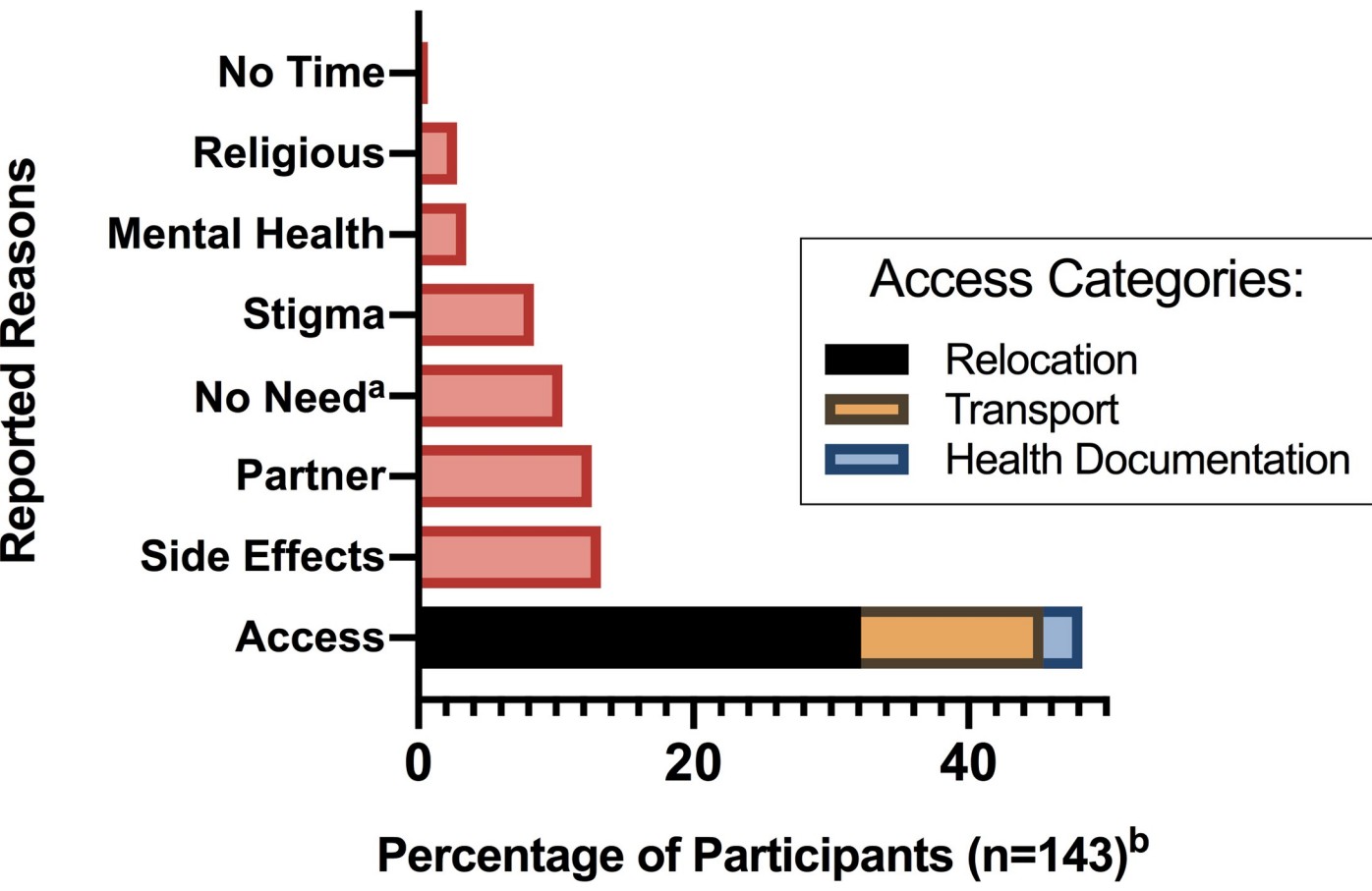

**Fig 1. Free response reasons reported by participants for art interruption.** [a]Participants expressed that they stopped taking ART because they felt healthy. [b]The free response reason for ART interruption was missing in 7 reinitiating participants.

findings are consistent with prior studies in Malawi and other countries with Option B+ programs that observe greater LTFU among PMTCT patients when compared with women who begin ART for their own health [6, 7, 18–22]. In our study, which reflects a more mature Option B+ program (2015–2019 versus 2011–2012) [6, 7, 19], we demonstrate that the challenges of sustained care engagement among women who initiate ART during pregnancy or breastfeeding persist.

Option B+ patients may face pressure from health care workers to initiate ART through test-and-treat policies, and being asymptomatic at treatment initiation may lead women to question the necessity of HIV care and particularly, of long-term ART [23]. Mothers have a strong motivation to limit transmission of HIV to their child [24, 25], but once MTCT is prevented, the barriers to remaining in care may impede ART continuation: LTFU is highest during the first year of Option B+ care compared to two or three years after initiation, revealing a vulnerable period postpartum [6]. Additionally, women who learn of their HIV status during antenatal care may have less opportunity to process a new HIV diagnosis and receive sufficient counseling, which may contribute to them not feeling prepared to start ART or disclose to their partner or relatives [25–27].

At the same time, in our reinitiating cohort, women started on ART for PMTCT rather than for their own health had a longer treatment duration prior to stopping ART. This observation may reflect a desire to remain on ART for the duration of pregnancy and breastfeeding

that then waned once a child is born or has finished breastfeeding. Future work can determine whether this difference in ART duration is a pattern observed in all women LTFU, as our study includes only those who are reinitiating care at a pregnancy. Ongoing and targeted counseling during the perinatal period may be required to clarify the need for lifelong ART treatment adherence and to improve sustainable engagement in care after delivery of a healthy infant.

Prior literature has reported that younger age [6, 7, 19, 28, 29] and lower education status [7, 18, 30] are associated with lower ART engagement. We found the same association in our sample. Older women and those with a higher level of completed education may have better mechanisms for accessing supplemental and ongoing sources of counseling, beyond what a busy healthcare worker can provide in government clinics. Prior interventions within Option B+ have noted the potential of 'mentor mother' programs, which pair newly diagnosed pregnant women with mothers who have prior experience with effective HIV health maintenance. Encouragingly, such peer-based support is associated with improved uptake in Option B + through both facility- and community-based peer support models [31]. Further investigation is needed to determine if social support programs specifically geared to younger mothers or those with limited educational completion can help improve long-term care engagement.

The role of relationship dynamics, as well as self- and partner disclosure patterns, have been less well studied among women living with HIV, yet may play a role in influencing long term engagement. We examined disclosure patterns in our cohort and found that non-disclosure of HIV status to their partner and lack of awareness of their partner's HIV status were both associated with a history of ART interruption. A lack of partner communication about HIV status may impede women's ability to remain in care. At Bwaila Hospital, where a majority of our participants were recruited, approximately 90% of women come to their first ANC visit without their partner and therefore must decide about whether to disclose their status to their partner if they are found to be HIV positive [32]. Fear of accusation of bringing HIV into the family, abandonment, loss of financial support, or violence [8, 33–36] may compromise long term care engagement for women diagnosed with HIV.

Male partner involvement initiatives could have a substantial impact for maintaining women in HIV care. Couples' HIV testing and counseling (CHTC) for example, which allows a couple to mutually disclose through a counselor in a clinic environment, is an effective strategy for supporting HIV status disclosure for women. CHTC interventions have led to increased ART uptake, care engagement, and decreased MTCT [37–39]. CHTC has been associated with improved social support indices not only from a woman's partner, but also family and peers one month after counseling [40]. Indeed, a combination of HIV status disclosure, supportive behaviors, and male ANC attendance was linked to improved maternal engagement at 18 months in Malawi [41]. Currently, healthcare workers in Malawi describe the setup of government ANC clinics as an impediment to male participation, with men deeming ANC a woman's space [42, 43]. Ongoing efforts to improve male participation include improving the male-friendliness of ANCs, home-based testing, secondary distribution of self-test kits, phone invitation, and physical tracing of male partners [32, 44–46]. Since most studies focus on ANC engagement and ART uptake or early follow-up, continued work on the long-term effects of partner involvement are warranted. Further interventions should encourage male partner involvement beyond ANC and CHTC, with a family-based approach encouraging couples' HIV visits (regardless of partner serostatus, a strategy currently being investigated by our group in Malawi, R00MH104154).

A majority of the women in our study did not intend their current pregnancy. The high level of unintended pregnancy in our study population mirrors that in a prior study in Malawi [47], reinforcing that many women living with HIV in Malawi desire to limit childbearing but

do not have access to effective contraceptive methods or family planning counseling. Pregnancy intent was associated with the reinitiating cohort in unadjusted analyses, but not in adjusted analyses. Attitudes towards reproduction in women living with HIV are complex given the experience of HIV-related stigma while in a cultural context that highly values childbearing [48, 49]. The lower prevalence of contraception use in the reinitiating cohort may reflect a lack of engagement with the health care system more broadly; in 2011, Malawi issued guidelines for provider-initiated family planning counseling in ART clinics for all patients 15 years or older [13]. In addition, women who do not feel comfortable disclosing their HIV status to their partner or eliciting their partner's HIV status may have less agency to access and advocate for family planning. Prior work in Malawi has demonstrated that targeting men for family planning interventions can increase contraceptive uptake [50]. Future work in family-based ART care can incorporate family planning counseling along with CHTC to determine whether partner involvement can promote both ART engagement and effective family planning.

According to participants in our study, access to care was the most common reason reported for ART interruption, with implications for access improvements. In particular, women cited challenges due to relocation. Women move often in sub-Saharan Africa; some women may move to be closer to family members during pregnancy [51], while others may follow husbands moving for job opportunities. Relocation often requires women to transfer clinics for ART care. In Malawi, changing ART clinics involves obtaining a transfer letter from the original clinic and reporting to the new one or testing in a new clinic. Due to the high volume of patients and short supply of staff in government clinics, there may be a limited opportunity for outlining this process to patients. Presenting to the clinic for a transfer letter may be a burden on many patients and women may also fear a negative response from healthcare workers should they desire to reinitiate ART after a gap in treatment. Systems-level interventions, such as the universal adoption of a centralized electronic medical record (EMR) monitoring system at all ART clinics in Malawi, would simplify the process of transferring clinics by obviating the need for transfer letters. Ongoing efforts utilizing biometric fingerprint scanning to uniquely identify patients at ART clinics can further improve the ease and monitoring of engagement in care [52]. In the meantime, health care workers educating women on initiating ART should include instructions for transferring HIV care, and demonstrate welcoming attitudes towards women transferring or returning to care as they constitute a sizeable percentage of ART reinitiators [7, 53].

Our study is limited by its cross-sectional nature, preventing causal conclusions about the temporality between predictors and the outcome of ART interruption. Variables at study initiation were largely self-reported, so there is the potential for error stemming from social desirability, misinterpretation, and recall bias. In-depth interviews of Malawian women living with HIV have previously identified the complex personal decision-making surrounding treatment decisions [54]. In the free-response portion of our study, the majority of women reported access to care as a barrier to ART engagement, however women may have been less likely to report more subjective reasons for treatment discontinuation given our study format. Our conclusions likely address external factors affecting ART engagement; future work on the influence of internal patient circumstances is warranted. In terms of generalizability, women in the reinitiating cohort were re-presenting for care so our population does not reflect women who are LTFU and have not reinitiated care. Future studies should explore factors associated with LTFU in a wider female population. Strengths of our study include assessing viral load and length of treatment where records were available, collecting quantitative and qualitative data to determine associations with ART interruption, and exploring partner relationship and disclosure factors contributing to an interruption in treatment.

## Conclusions

Amongst Malawian women initiating antenatal care, we identified that beginning ART for PMTCT, younger age, lower education level, lack of HIV status disclosure to a partner, lack of awareness of a partner's HIV status, and not using contraception were associated with ART interruption. Women cited access difficulties, particularly relocation, as the most common reason they stopped HIV care. Understanding barriers and challenges for participants in a maturing Option + program can help inform the design and evaluation of interventions to achieve the elimination of MTCT agenda. Our results have several implications for interventions that could improve women's long-term engagement in HIV care. Future interventions could include counseling on the need for lifelong ART, implementing a family-based model of HIV care promoting male partner involvement, and streamlining the clinic transfer process.

## Supporting information

**S1 File.**
(PDF)

## Acknowledgments

We gratefully acknowledge all of the women who participated in this study. Many thanks to all additional members of the S4 research team: Chimwemwe Baluwa, Gabriel Banda, Maganizo Chagomerana, Linda Chikopa, Limbikani Chimndozi, Alvis Dauya, Ntchindi Gondwe, Mathias John, Bridget Kafuwa, Jane Kilembe, Phaleda Kumwenda, Wiza Kumwenda, Chalimba Lusewa, Don Makonokaya, Clement Mapanje, Albans Msika, Kingsley Msimuko, Brian Mthiko, Christopher Mwafulirwa, Dan Nansongole, Macneil Ngongondo, Juliana Ngwira, Trywin Phiri, Emmanuel Singogo, Anthony Stambuli, Terence Tafatatha, Gerald Tegha, Anne Thom, Austin Wesevich, and Agness Zilore.

## Author Contributions

**Conceptualization:** Simone A. Sasse, Bryna J. Harrington, Madalitso Maliwichi, Allan N. Jumbe, Irving F. Hoffman, Nora E. Rosenberg, Jennifer H. Tang, Mina C. Hosseinipour.

**Data curation:** Simone A. Sasse, Bryna J. Harrington, Bethany L. DiPrete, Shaphil D. Wallie, Allan N. Jumbe, Mina C. Hosseinipour.

**Formal analysis:** Simone A. Sasse, Bryna J. Harrington, Bethany L. DiPrete, Maganizo B. Chagomerana, Nora E. Rosenberg, Jennifer H. Tang, Mina C. Hosseinipour.

**Funding acquisition:** Simone A. Sasse, Bryna J. Harrington, Mina C. Hosseinipour.

**Investigation:** Allan N. Jumbe, Irving F. Hoffman, Mina C. Hosseinipour.

**Methodology:** Bryna J. Harrington, Bethany L. DiPrete, Laura Limarzi Klyn, Madalitso Maliwichi, Allan N. Jumbe, Irving F. Hoffman, Nora E. Rosenberg, Jennifer H. Tang, Mina C. Hosseinipour.

**Project administration:** Shaphil D. Wallie, Allan N. Jumbe, Mina C. Hosseinipour.

**Software:** Shaphil D. Wallie.

**Supervision:** Nora E. Rosenberg, Jennifer H. Tang, Mina C. Hosseinipour.

**Writing – original draft:** Simone A. Sasse.

**Writing – review & editing:** Simone A. Sasse, Bryna J. Harrington, Bethany L. DiPrete, Maganizo B. Chagomerana, Laura Limarzi Klyn, Madalitso Maliwichi, Irving F. Hoffman, Nora E. Rosenberg, Jennifer H. Tang, Mina C. Hosseinipour.

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
