## [Decision Letter · Decision Letter 0]

11 Feb 2022

PONE-D-21-07266Factors associated with a history of loss to follow-up among pregnant women living with HIV in Malawi: A cross-sectional studyPLOS ONE

Dear Dr. Sasse,

Thank you for submitting your manuscript to PLOS ONE. After careful consideration, we feel that it has merit but does not fully meet PLOS ONE’s publication criteria as it currently stands. Therefore, we invite you to submit a revised version of the manuscript that addresses the points raised during the review process.

The manuscript has been evaluated by three reviewers, and their comments are available below.

The reviewers carefully assessed the findings of this manuscript, and while they expressed interest in the findings of the study, they have raised a number of concerns. Of particular note, reviewers raised concerns about the appropriateness of the statistical methods used.

Could you please carefully revise the manuscript to address and respond to all of the comments raised? We would recommend that you consult a statistician to review this manuscript given the particular concerns raised.

We look forward to receiving your revised manuscript.

Kind regards,

Jamie Royle

Academic Editor

PLOS ONE

Journal Requirements:

2. Please include additional information regarding the survey or questionnaire used in the study and ensure that you have provided sufficient details that others could replicate the analyses. For instance, if you developed a questionnaire as part of this study and it is not under a copyright more restrictive than CC-BY, please include a copy, in both the original language and English, as Supporting Information.  If the original language is written in non-Latin characters, for example Amharic, Chinese, or Korean, please use a file format that ensures these characters are visible.

3. Please state whether you validated the questionnaire prior to testing on study participants. Please provide details regarding the validation group within the methods section.

Reviewers' comments:

Reviewer's Responses to Questions

**Comments to the Author**

1. Is the manuscript technically sound, and do the data support the conclusions?

Reviewer #1: Yes

Reviewer #2: Yes

Reviewer #3: Yes

2. Has the statistical analysis been performed appropriately and rigorously? 

Reviewer #1: Yes

Reviewer #2: No

Reviewer #3: I Don't Know

3. Have the authors made all data underlying the findings in their manuscript fully available?

Reviewer #1: Yes

Reviewer #2: Yes

Reviewer #3: Yes

4. Is the manuscript presented in an intelligible fashion and written in standard English?

Reviewer #1: Yes

Reviewer #2: Yes

Reviewer #3: Yes

5. Review Comments to the Author

Reviewer #1: Review of manuscript “Factors associated with a history of loss to follow-up among pregnant women living”.

Sasse and colleagues did a cross-sectional study of factors associated with ART interruption among pregnant women living with HIV in Malawi using baseline data from an observational study to evaluate the safety and efficacy of Malawi’s PMTCT program. The manuscript is well written and adheres to reporting guidelines. Statistical and qualitative data analyses are methodologically sound. The conclusions of the study are supported by data presented in the manuscript. I have some relatively minor suggestions for improvement.

1. The authors included women retained on ART and those returning to care after an interruption. Women lost to follow-up who did not return to care are not included in the study. While the authors acknowledge that results cannot be generalized to “women who are LTFU and have not reinitiated care” (line 376), they sometimes use language that suggests otherwise. For example, in line 91, the authors state that the aim of the study is “[t]o determine factors associated with LTFU in a more mature Option B+…”. Along similar lines, the expression “history of loss to follow-up” does not necessarily imply the women returned to care. I suggest that the authors use the expression “interruption of ART” or “representation to HIV care” instead of “history of loss to follow-up” and “loss to follow-up” throughout the manuscript to clarify that this is an analysis of factors associated with interruption of ART and not an analysis of factors associated with LTFU.

2. Qualitative data on reasons for treatment interruptions were collected by study nurses at the clinics. The authors should consider expanding the discussion on how this interview situation may have influenced the reported reasons for treatment interruptions. Are women who returned to care after a treatment interruption who “fear a negative response from healthcare workers should they desire to reinitiate ART” able to respond honestly to this question? Most women reported external reasons for interrupting ART. The interview situation may make external attribution and overreporting of factors outside of the control of the individual likely. In-depth interviews of pregnant women who discontinued ART revealed subjective reasons for treatment interruptions (10.1016/j.socscimed.2016.04.013). A more balanced discussion may be warranted. This more general meta-review may also be helpful: 10.1016/j.socscimed.2004.11.063).

3. Please clarify why criteria 3 applies to confirm that women had not been on ART “3) documentation of HIV-positive status while pregnant or breastfeeding after July 2011 (when the implementation of Option B+ began)” line 112.

Thanks for the opportunity to review this paper.

Best regards,

Andreas Haas

Reviewer #2: Overall comments:

1. Very interesting reading, with some important conclusions. However -

2. The paper needs 'polishing' by a senior researcher

3. The use of Poisson regression for this analysis needs to be reconsidered. This is a method typically used when the dependent variable is a count, and one or more of the covariates is continuous or ordinal. By changing all variables to binary (including the 'outcome' of a history of LTFU, you're violating the first assumption of Poisson regression. Suggest considering revising methods to utilize either logistic regression (given your binary outcome). You may also want to reconsider whether to utilize categorical co-variates instead of just binary.

- If you believe you've chosen the best statistical methods, please provide a brief justification in your methods section.

4. More description around the quantification of your qualitative responses to interviews need to be added to the methods

5. The discussion section is very long - consider focusing on 1-2 key takeaway messages.

Reviewer #3: This is a concisely written paper that shows some health system challenges in the management of clients under the Option B+ approach. A few areas to bring out clarity of the paper:

Explain how the cohort that was reinitiating care was identified- by whom and what had brought them back to the facility. Were they traced or they presented at freewill?

Some details about the reinitiating cohort would help to contextualise the study such as, whether a woman's child was alive and well, presence of a repeat pregnancy and anything that may describe them more-- were they originally enrolled at the same facility?

How were the study results influenced by the settings of the study? Are the two sites comparable?

The qualitative analysis- There is need for more clarity so that a qualitative researcher should not be misled thinking that this has a qualitative component rather it had qualitative responses that were organised using a thematic approach and then frequencies were tabulated. Leaving out some of that detail almost attracts questions about how quality was ensued for the qualitative responses and the need to see quotes.

6. PLOS authors have the option to publish the peer review history of their article (what does this mean?). If published, this will include your full peer review and any attached files.

Reviewer #1: **Yes: **Andreas D Haas

Reviewer #2: No

Reviewer #3: **Yes: **Alinane Linda Nyondo-Mipando

---

## [Author Response · Author response to Decision Letter 0]

27 Mar 2022

Reviewer #1 

1. Thank you for your suggestion. We agree that clarification of the study population was warranted and changed the language throughout the manuscript to reflect that we studied women reinitiating care as opposed to women truly lost to follow up. We have made this change in the title: “Factors associated with a history of treatment interruption among pregnant women living with HIV in Malawi: A cross sectional study,” the abstract (ex/ line 40, 43), and the main text (ex/ line 102). 

2. We thank the reviewer for the recommendation of a more nuanced discussion of free response reasons for ART discontinuation. We agree that given the interview setting within an antenatal clinic, respondents could be more likely to report external factors beyond their control. We have included this discussion in lines 423-431, and included the reference suggested: 

Zhou A. The uncertainty of treatment: Women’s use of HIV treatment as prevention in Malawi. Soc Sci Med. 2016;158:52–60.

3. “Documentation of HIV-positive status while pregnant or breastfeeding after July 2011 (when the implementation of Option B+ began)” was part of inclusion criteria to capture women likely offered ART but unwilling to disclose that they had interrupted therapy due to high uptake of ART. However, no women in the study fit this criterion. 

Reviewer #2 

1,2. Thank you for your suggestion - three senior researchers included in the co-authors have reviewed the paper for readability. 

3. We thank the reviewer for raising these important points and believe we have chosen the most appropriate statistical methods. Our primary outcome of interest is a binary outcome (LTFU). When a binary outcome is being estimated, either a logistic regression model (to estimate an odds ratio) or a log-binomial regression model (to estimate a prevalence ratio) is typically used. However, sometimes, a log-binomial regression model does not converge. In these instances, a modified Poisson regression model can be used to approximate a prevalence ratio. We prefer a prevalence ratio to an odds ratio due to ease of interpretation. We provide the following citations explaining the mathematical underpinnings in greater detail: 

14. Zou G. A Modified Poisson Regression Approach to Prospective Studies with Binary Data. Am J Epidemiol. 2004;159(7):702–6. 

15. Barros A, Hirakata V. Alternatives for logistic regression in cross-sectional studies: an empirical comparison of models that directly estimate the prevalence ratio. BMC Med Res Methodol. 2003;3(21):16–21. 

We have added the word “modified” in the abstract and text for greater clarity (line 48, 170, 172). In the text we explain this decision in the following way:

“To explore potential factors associated with LTFU, we used modified Poisson regression models with robust variance to calculate unadjusted prevalence ratios (PR). Modified Poisson regression with robust variance estimates were utilized due to issues with convergence of log-binomial regression models (14).”

4. We have clarified our process of quantification of qualitative responses in the methods section in lines 181-193, as also suggested by Reviewer #3. The description now reads as, “Qualitative free responses for ART discontinuation in the reinitiating cohort were organized using thematic content analysis (16). One researcher reviewed all participant responses to the question and developed eight categories for coding. Two independent researchers applied the codebook to the responses. The initial inter-rater reliability Cohen’s kappa was 0.86. Discrepancies were reviewed and resolved by consensus. The frequency of reason for ART discontinuation category was tabulated and reported.” 

5. We were unable to substantially modify the length of the discussion due to suggestions by other reviewers. However, we take note the impression of Reviewer 3 who felt the manuscript was concise. 

Reviewer #3 

Thank you for your comments helpful for improved contextualization of our study. All of the patients presented at free will to an antenatal clinic and were subsequently approached (no tracing occurred)– we have provided clarification in lines 113-115 with the following description, “For this study, women who presented for an antenatal clinic visit and tested positive for HIV during any trimester of pregnancy were approached for recruitment into either: 1) the retained cohort, or 2) the reinitiating cohort.” To better address how the study results may have been influenced by the setting of the study, we have included lines 423-431 as described for Reviewer 1’s second point above. We have additionally clarified that qualitative responses were organized using a thematic approach and the frequencies were tabulated, as described in our response to Reviewer 2’s point 4. We unfortunately did not collect information on the prior location of ART provision in the reinitiating cohort, so we cannot provide this additional contextualization. The two sites, Area 18 Health center and Bwaila Hospital, are comparable in providing comprehensive antenatal and HIV care for patients. However, Area 18 is smaller and less busy.

---

## [Editor Report · Decision Letter 1]

4 Apr 2022

Factors associated with a history of treatment interruption among pregnant women living with HIV in Malawi: A cross-sectional study

PONE-D-21-07266R1

Dear Dr. Sasse,

We’re pleased to inform you that your manuscript has been judged scientifically suitable for publication and will be formally accepted for publication once it meets all outstanding technical requirements.

Finally, I would like to disclose that I participated as a reviewer for the initial evaluation of this manuscript. PLOS ONE subsequently invited me to act as Guest Academic Editor and decide whether this manuscript is suitable for publication.

Kind regards,

Andreas D Haas, PhD

Guest Editor

PLOS ONE
---

## [Editor Report · Acceptance letter]

11 Apr 2022

PONE-D-21-07266R1 

Factors associated with a history of treatment interruption among pregnant women living with HIV in Malawi: A cross-sectional study  

Dear Dr. Sasse:

I'm pleased to inform you that your manuscript has been deemed suitable for publication in PLOS ONE. Congratulations! Your manuscript is now with our production department. 

Kind regards, 

on behalf of

Dr. Andreas D Haas 

Guest Editor

PLOS ONE